# Effect of the Interaction between Elevated Carbon Dioxide and Iron Limitation on Proteomic Profiling of Soybean

**DOI:** 10.3390/ijms232113632

**Published:** 2022-11-07

**Authors:** José C. Soares, Hugo Osório, Manuela Pintado, Marta W. Vasconcelos

**Affiliations:** 1CBQF—Centro de Biotecnologia e Química Fina—Laboratório Associado, Escola Superior de Biotecnologia, Universidade Católica Portuguesa, Rua Diogo Botelho 1327, 4169-005 Porto, Portugal; 2i3S—Instituto de Investigação e Inovação em Saúde, Universidade do Porto, 4200-135 Porto, Portugal; 3Ipatimup—Institute of Molecular Pathology and Immunology of the University of Porto, University of Porto, 4200-135 Porto, Portugal

**Keywords:** elevated CO_2_, iron limitation, soybean, proteomic profiling

## Abstract

Elevated atmospheric CO_2_ (eCO_2_) and iron (Fe) availability are important factors affecting plant growth that may impact the proteomic profile of crop plants. In this study, soybean plants treated under Fe-limited (0.5 mM) and Fe-sufficient (20 mM) conditions were grown at ambient (400 μmol mol^−1^) and eCO_2_ (800 μmol mol^−1^) in hydroponic solutions. Elevated CO_2_ increased biomass from 2.14 to 3.14 g plant^−1^ and from 1.18 to 2.91 g plant^−1^ under Fe-sufficient and Fe-limited conditions, respectively, but did not affect leaf photosynthesis. Sugar concentration increased from 10.92 to 26.17 μmol g FW^−1^ in roots of Fe-sufficient plants and from 8.75 to 19.89 μmol g FW^−1^ of Fe-limited plants after exposure to eCO_2_. In leaves, sugar concentration increased from 33.62 to 52.22 μmol g FW^−1^ and from 34.80 to 46.70 μmol g FW^−1^ in Fe-sufficient and Fe-limited conditions, respectively, under eCO_2_. However, Fe-limitation decreases photosynthesis and biomass. Pathway enrichment analysis showed that cell wall organization, glutathione metabolism, photosynthesis, stress-related proteins, and biosynthesis of secondary compounds changed in root tissues to cope with Fe-stress. Moreover, under eCO_2_, at sufficient or limited Fe supply, it was shown an increase in the abundance of proteins involved in glycolysis, starch and sucrose metabolism, biosynthesis of plant hormones gibberellins, and decreased levels of protein biosynthesis. Our results revealed that proteins and metabolic pathways related to Fe-limitation changed the effects of eCO_2_ and negatively impacted soybean production.

## 1. Introduction

Atmospheric carbon dioxide (CO_2_) concentrations are projected to increase to at least 700–1000 μmol mol^−1^ at the end of this century [1]. The trend in crop responses under elevated CO_2_ (eCO_2_) is supposed to have a noticeable influence on the global food chain and may threaten human nutrition [2]. The impact of eCO_2_ varies among different crop species, but eCO_2_ generally improves the photosynthesis of the C3 plants [3] by repressing the oxygenase activity of ribulose-1,5-bisphosphate and by improving carbon assimilation used for plant growth [4]. Nevertheless, plant-based responses to eCO_2_ are affected by nutrient bioavailability, including iron (Fe) [5]. Iron is considered a limiting factor and plays a key role in biomass production and seed yield in crops including tomato [5], spinach [6], potato, and rice [7]. The mineral element is of particular significance because of its role in photosynthetic CO_2_ fixation, which utilizes Fe as a crucial element to ensure photosynthetic efficiency [4]. Anyway, Fe is the most frequently deficient micronutrient in the human diet, affecting an estimated 2 billion people [8]. While eCO_2_ promotes plant growth, the interaction with Fe-limitation has not been well documented. Due to the increase in plant growth, the plant requirement for nutrients also increases, and the restriction of macro and micronutrient content at eCO_2_ generally inhibits the increase in plant biomass [4]. Accordingly, low nitrate supply limits shoot growth and hormonal responses to eCO_2_ [2], involving alterations in protein synthesis and metabolic pathways associated with eCO_2_ and nitrate. The proteomic analysis demonstrated an increase in the expression of many proteins due to eCO_2_ under adequate nitrate levels involved with cell cycle and proliferation, transcription and translation, photosynthesis, amino acids synthesis, sucrose, and starch metabolism, and ABA signaling pathways. Proteomic analysis showed that eCO_2_ affected various metabolic processes and pathways, such as photosynthetic carbon fixation, respiratory metabolism, cellular growth, and stress defense [9,10]. Yu et al. [11] also demonstrated that eCO_2_ improved heat tolerance in bermudagrass attributed to metabolic pathways during which proteins and metabolites were upregulated, including light reaction and carbon fixation of photosynthesis, glycolysis and TCA cycle, and amino acid metabolism.

Iron is an essential micronutrient that impacts plant productivity, and its bioavailability to plants is often limited in calcareous soils [12], which account for ~30% of the world’s agricultural soil. In tomato plants grown in Fe-limited and Fe-sufficient conditions, eCO_2_ increased plant biomass and root-to-shoot ratio compared to plants grown in ambient CO_2_ (aCO_2_) conditions. The percentage increase in biomass was higher in Fe-limited plants compared to Fe-sufficient plants. Therefore, shoot fresh weight increased under eCO_2_ conditions by 22% and 44% in Fe-sufficient and Fe-limited conditions, respectively, and fresh root weight by 43% and 97% [5]. Likewise, eCO_2_ induced biomass production in barley grown at Fe-sufficient and Fe-deficient conditions and increased shoot Fe concentration and Fe-acquisition mechanisms in Fe-deficient plants, indicating an improved internal Fe utilization [13]. The proteomic characterization of Fe deficiency responses in cucumber roots revealed that most of the increased proteins belong to glycolysis and nitrogen metabolism, and proteins with low expression levels were related to the metabolism of sucrose, structural carbohydrates, and proteins [14]. In addition, proteins associated with stress adaptation, reactive oxygen species-related proteins, and mitochondrial proteins were differentially expressed under long-term Fe-restriction on the roots of different pea cultivars [15]. López-Millán et al. [16] revealed some common elements in proteome under Fe-limitation involving several plant species. Oxidative stress and defense-related proteins, C and N metabolism-related proteins, cell wall proteins, secondary metabolism-associated proteins, energy-related proteins, transport, and protein metabolism have been identified as differentially accumulated proteins among plant species.

Therefore, we consider that it is relevant to discover the complex response mechanisms of soybean plants to future climatic conditions, such as eCO_2_ and Fe-limitation. This study aimed to understand how soybean plants respond to eCO_2_, Fe-stress, and their combination at the level of plant growth, carbohydrate content, photosynthesis, and the expression profile of the leaf and root proteome. At the same time, it was our purpose to investigate if eCO_2_ can mitigate the adverse effects of Fe-limitation in soybean plants and provide several insights into improving stress tolerance in the future environment.

## 2. Results

### 2.1. Interactive Effects of eCO_2_ and Fe-Limitation on Plant Biomass

Both eCO_2_ and Fe-limitation showed significant effects (*p* < 0.05) on plant biomass with a significant CO_2_ x Fe interaction (Table 1). Elevated CO_2_ increased plant dry weight from 2.14 to 3.14 g plant^−1^ under Fe-sufficient conditions and from 1.18 to 2.91 g plant^−1^ under Fe-limited conditions. Furthermore, Fe-limitation significantly decreased plant biomass at aCO_2_ (*p* < 0.05) but not at eCO_2_ (*p* > 0.05).

### 2.2. Interactive Effects of eCO_2_ and Fe-Limitation on Photosynthesis Parameters

Fe-limitation showed a significant effect on Pn (*p* < 0.05), whereas at eCO_2_ (*p* > 0.05, Table 1), a significant effect was not detected. The decrease in Pn due to Fe-limitation was −21.7% and −28.4% under eCO_2_ and aCO_2_ conditions, respectively. Elevated CO_2_ significantly decreased gs under both Fe supplies (*p* < 0.05). Moreover, under Fe-limitation, gs was reduced at aCO_2_ (*p* < 0.05) and at eCO_2_ conditions (*p* > 0.05). The transpiration rate (Tr) was decreased by eCO_2_ (*p* < 0.05) and by Fe-stress at aCO_2_ conditions (*p* < 0.05).

### 2.3. Interactive Effects of eCO_2_ and Fe-Limitation on Sugar Content

Elevated CO_2_ showed a significant effect on sugar concentration in the root (*p* < 0.05, Table 1) and leaf tissues (*p* < 0.05) but was not affected by Fe-limitation (*p* > 0.05). Sugar concentration increased from 10.92 to 26.17 μmol g FW^−1^ in the roots of Fe-sufficient plants and from 8.75 to 19.89 μmol g FW^−1^ in the roots of Fe-limited plants after exposure to eCO_2_. In leaves, sugar concentration increased from 33.62 to 52.22 μmol g FW^−1^ in Fe-sufficient conditions and from 34.80 to 46.70 μmol g FW^−1^ in Fe-limited conditions.

### 2.4. Functional Categories of Differentially Expressed Proteins

Appendix A go into depth about the entire list of proteins that were found and measured. Proteins with a *p*-value below 0.05 and a fold change ratio above 1.2 or below 1/1.2 were considered as DEPs (Figure 1 and Appendix A).

Among the identified proteins, 705 were differentially expressed in root tissues considering a fold change ratio of above 1.2 or below 1/1.2 (*p* < 0.05, Figure 1A). We found 27 and 136 proteins that were upregulated under eCO_2_ (Fe+ELE/Fe+AMB) and Fe-limitation (Fe-AMB/Fe+AMB), respectively, with an overlap of one protein (Fe2OG dioxygenase domain-containing protein, Appendix A). Furthermore, 167 and 110 proteins were downregulated by eCO_2_ and Fe-limitation, respectively, with an overlap of two HMA domain-containing proteins, one superoxide dismutase and one 2-isopropylmalate synthase. Compared to control plants, 141 proteins were up and 79 downregulated by the interaction of eCO_2_ with Fe-limitation (Fe-ELE/Fe+AMB). We also found 139 up and 119 downregulated proteins in Fe-ELE/Fe-AMB conditions. Protein expression profiles could be correctly distinguished between treatments using the heat map of differentially expressed proteins (Figure 1B). The metabolism overview maps showed changes at the protein level obtained using the MapMan software. Proteins associated with redox regulation and cell wall metabolism decreased the expression levels under eCO_2_, and photosynthetic light reactions proteins were upregulated by Fe-limitation, as shown in Figure 2.

In addition, proteins related to photosynthetic light reactions, glycolysis, and redox regulation were upregulated in Fe-ELE/Fe+AMB conditions. Enzymes associated with secondary metabolism, glycolysis, and redox regulation were upregulated in Fe-ELE/Fe-AMB conditions. The number of differentially regulated proteins, according to functional categories and cellular localization, is summarized in Figure 3.

Most enzymes in root tissues with assigned functions were “protein-related,” “stress-related,” “secondary metabolism-related,” and “RNA-related,” and were mostly downregulated in response to eCO_2_. Protein synthesis, photosynthesis-related, and redox-related enzymes were mainly upregulated under Fe-limited conditions. In Fe-ELE/Fe+AMB conditions, most proteins were associated with photosynthesis, hormone metabolism, redox regulation, signaling, and glycolysis and upregulated. In addition, most enzymes related to protein synthesis were downregulated, and enzymes associated with secondary metabolism, hormone metabolism, stress, and glycolysis were upregulated in the Fe-ELE treatment compared with Fe-AMB. According to their cellular location, most proteins identified were in the cytosol, followed by the extracellular space and nucleus.

In leaves, a total of 589 proteins were differentially expressed (fold change ratio > 1.2 or <1/1.2 and *p* < 0.05). As shown in Figure 4, the gene expression profile between treatments could be distinguished and included 72 and 101 proteins that were up and downregulated, respectively, under eCO_2_.

However, Fe-limitation resulted in 146 up and 104 downregulated proteins. In upregulated, an overlap of 14 proteins (Appendix A), whereas, in downregulated proteins, an overlap of 27 proteins was observed between eCO_2_ and Fe-limitation. The interaction of Fe-limitation and eCO_2_ resulted in 83 and 76 proteins being upregulated and downregulated, respectively. From the upregulated proteins, the interaction induced an overlap of 28 proteins with Fe-limitation or 27 proteins with eCO_2_. Regarding downregulated proteins, plants grown under Fe-limitation and eCO_2_ had an overlap of 41 proteins with Fe-stress and 36 proteins with eCO_2_. Furthermore, 74 and 136 proteins were upregulated and downregulated in the Fe-ELE/Fe-AMB conditions. Figure 5 displays the metabolism overview maps in the leaf tissues, and eCO_2_ increased the expression of enzymes involved in glycolysis, photosynthesis, and redox homeostasis, while Fe-limitation decreased the expression of proteins involved in photosynthetic light processes.

In Fe-ELE/Fe+AMB conditions, the expression of proteins involved in glycolysis and redox homeostasis increased. However, in Fe-ELE/Fe-AMB, enzymes related to glycolysis and carbohydrate metabolism were increased. The number of DEPs, according to functional categories and cellular compartments, is described in Figure 6. Elevated CO_2_ increases the proteins involved in stress, photosynthesis, and glycolysis while downregulating the enzymes involved in signaling and protein synthesis. Furthermore, protein synthesis and secondary metabolism-related proteins exhibited high expression levels, while photosynthesis-related proteins had low levels under Fe-limitation. Most glycolysis-related proteins were downregulated in Fe-ELE/Fe+AMB. In addition, most proteins associated with carbohydrate metabolism, photosynthesis, stress, and glycolysis were upregulated, and protein synthesis-related enzymes were downregulated in Fe-ELE compared with Fe-AMB. Concerning their cellular location, most of the proteins identified were in the cytosol, followed by the plastid and extracellular space (Figure 6).

### 2.5. Metabolic Pathways Related to the Interaction of eCO_2_ and Fe-Limitation

To gain a better understanding of the DEPs between different treatments, pathway enrichment analysis was conducted and shown in Table 2 and Table 3. The photosynthesis-related pathways (bin 1) were upregulated in the Fe-AMB vs. Fe+AMB and Fe-ELE vs. Fe+AMB treatments in the root. Glycolysis (bin 4.1) was not affected by eCO_2_ and Fe-limitation but was upregulated under Fe-ELE treatment compared to Fe+AMB and Fe-AMB. Cell wall organization (bin 10) and metal binding (bin 15.2) were not affected by eCO_2_ but downregulated by Fe-limitation and the interaction of eCO_2_ and Fe-limitation. The secondary metabolism (bin 16) was negatively downregulated by Fe-limitation and upregulated by eCO_2_ under Fe-limitation (Fe-ELE vs. Fe-AMB). Enzymes involved in hormone metabolism (bin 17.6) had a higher level of expression at eCO_2_ under Fe-sufficient and Fe-limited conditions. Proteins associated with stress responses (bins 20.1 and 20.2.1) had low expression under Fe-stress. Moreover, an increase in ascorbate and glutathione metabolism (bin 21.2.1.3), which is an effective mechanism of plant detoxification, was induced by Fe-ELE compared with Fe+AMB conditions. Peroxidases (bin 26.12), phosphatases (bin 26.13), and oxygenases (bin 26.14) were not affected by CO_2_ enrichment but downregulated by Fe-limitation. The expression of glutathione S-transferase proteins (bin 26.9) was higher in Fe-limited plants. Inhibition of gene expression in RNA (bin 27) and protein synthesis (bin 29) pathways occurred under eCO_2_.

The expression levels of transport proteins (bin 34) involved in the translocation of solutes across membranes were not affected by eCO_2_. Regarding leaf tissues, listed photosynthesis (bin 1) associated pathways were downregulated by Fe-stress. Upregulation of sucrose and starch biosynthesis (bin 2.1) genes occurred at eCO2 under Fe-limited conditions, while downregulation was noticeable in low-Fe supply. Furthermore, glycolysis (bin 4.1) involved in the breakdown of sugars was upregulated by eCO_2_ under Fe-limited and Fe-sufficient conditions and not affected by Fe-stress. Stimulation of fatty acid degradation was apparent by eCO2 under Fe-limited conditions. Flavonoid biosynthesis (bin 16.8) was induced by Fe-stress and persisted unchanged under eCO_2_ conditions. Proteins related to stress response (bin 20.1) were upregulated by eCO_2_, Fe-limitation, and interaction of eCO_2_ with Fe-limitation. Furthermore, several heat-shock proteins (bin 20.2.1) were downregulated by Fe-limitation. Glutathione S-transferases (bin 26.9) and peroxidases (bin 26.12) were upregulated by Fe-limitation and not affected by eCO_2_. In addition, several enzymes involved in protein synthesis (bin 29) had lower expression levels at eCO_2_ under Fe-sufficient and Fe-limited conditions. Cysteine protease degradation (bin 29.5.3) was upregulated by Fe-limitation and by the interaction of eCO_2_ and Fe-limitation.

## 3. Discussion

To cope with Fe-stress and eCO_2_, soybean plants have evolved complex signaling and metabolic processes at the cellular, organ, and whole-plant levels. Elevated CO_2_ promoted plant growth under Fe-sufficient and Fe-limited conditions, as shown in Table 1. Similarly, this “fertilization effect” was reported in tomato [5] and barley plants [13] grown under low levels of Fe supply and eCO_2_. However, plant growth decreased under Fe-limitation, particularly at aCO_2_, suggesting that eCO_2_ could mitigate the Fe deficiency responses. Moreover, Pn was not affected by eCO_2_ but reduced in Fe-limited plants promoting leaf chlorosis (Table 1). The statistical analysis also revealed a significant interaction between eCO_2_ and Fe-stress on plant growth but not in Pn. Elevated CO_2_ significantly increased the sugar content in root and leaf tissues (Table 1). At eCO_2_, plants might surpass what they are capable of, using or distributing to sinks, increasing the carbohydrate content and possibly leading to feedback inhibition of photosynthesis [17]. Sugars are recognized to crosstalk with hormones acting on gene regulation and therefore modify nutrient uptake and transport, among other functions [17]. Lin et al. [18] suggested that sucrose acts as a signaling molecule, causing an increase in auxin and a subsequent increase in nitric oxide, leading to the FIT-mediated transcriptional regulation of FRO2 and IRT1 genes inducing Fe-uptake mechanisms. Although physiological aspects of soybean responses to eCO_2_ or Fe-limitation are well studied, proteomic profiling helps understand the molecular basis of soybean adaptation to the upcoming changing climate. Thus, several DEPs were found, in root and leaf tissues, due to eCO_2_ and Fe-limitation (Figure 1 and Figure 4). Exposure to eCO_2_ and Fe-stress leads to changes in the expression of genes involved in the carbon metabolism in soybean seedlings, especially the expression of genes related to glycolysis, starch and sucrose metabolism, and Myo-inositol metabolism. Elevated CO_2_ under Fe-limited conditions stimulated starch and sucrose biosynthesis. Therefore, protein abundance of most enzymes involved in starch (e.g., starch synthase, granule-bound starch synthase, starch-branching enzyme, and 1,4-alpha-glucan branching enzyme) and sucrose biosynthesis (e.g., sucrose-phosphate synthase), were upregulated by eCO_2_ conditions (Table 3). The central role of glycolysis is to break down glucose, produce ATP and generate precursors such as fatty acids and amino acids for anabolism [19]. Enzymes involved in glycolysis, such as enolase, pyruvate kinase, phosphofructokinase, glyceraldehyde-3-phosphate dehydrogenase, and phosphoglucomutase, were upregulated by eCO_2_ under Fe-limited conditions (Appendix A). Together, these findings led us to suppose that increased energy production via carbohydrate metabolism maintains energy homeostasis. Therefore, eCO_2_ increased the transcript levels of genes encoding enzymes involved in foliar cellular respiration, suggesting an improved flux through glycolysis driven by higher carbohydrate bioavailability (starch and sucrose) at eCO_2_. Ainsworth et al. [20] also found that growth at eCO_2_ led to the stimulation of foliar respiration in soybean plants. This adaptive balance enables plant growth even in the case of Fe-limitation.

In the present study, we found evidence of photosynthesis-related genes in roots (Table 2). DEPs enriched in photosynthesis were upregulated by Fe-limitation and by the interaction of eCO_2_ with Fe-limitation. Similarly, Kobayashi et al. [21] reported increased levels of transcription factors responsible for the coordinated expression of genes in chloroplast biogenesis in the roots of *Arabidopsis thaliana*. Accumulation of two golden-2-like transcription factors known to improve phototrophic performance and increase photosynthesis-related proteins in wheat roots under Fe deficiency was reported by Kaur et al. [22]. We hypothesize that this might be a strategy to cope with Fe-stress, thereby increasing CO_2_ fixation and demonstrating the possibility of root photosynthesis to improve plants’ carbon utilization. In leaves, consistent with protein expression levels, Fe-limitation depressed leaf photosynthesis, which agrees with other studies [4,23,24,25]. The Fe-AMB treatment produced a higher reduction in plant biomass when compared to the Fe-ELE treatment (Table 1). These results indicated that eCO_2_, particularly under Fe-limitation, increased the accumulation of photoassimilates due to eCO_2_-induced sugar metabolic pathways. This process is stimulated during adaptation to eCO_2_ to generate energy used to maintain leaf growth.

We found many genes involved in flavonoid biosynthesis upregulated after exposure to Fe-stress in leaves and some downregulated in soybean roots. The modulation of genes involved in the biosynthesis of flavonoids suggests that secondary metabolism also plays a role in Fe-stress responses. Several studies reported the effect of various stresses on secondary metabolism in plants [26,27]. Flavonoids are secondary plant products that are biologically active and perform different functions in plants as defense mechanisms against abiotic and biotic stresses [28]. Ahmed et al. [29] demonstrated that drought stress induced the expression of flavonoid biosynthesis genes in hybrid poplar plants and increased the accumulation of phenolic and flavonoid compounds with antioxidant activity. Iron deficiency also increased the expression of crucial enzymes in the flavonoid pathway in *Arabidopsis thaliana* roots [30]. Anyway, the emission of phenolic complexes into the rhizosphere, involved in Fe deficiency-induced responses, is considered a component of the strategy I plants. It was demonstrated in Arabidopsis roots that coumarins participate in Fe-chelation under Fe deficiency [31]. We found overexpression of feruloyl-CoA 6′-hydroxylase and coumarin synthase involved in coumarin biosynthesis under Fe-limitation in root tissues (Appendix A). The above results reinforced the role of coumarins in plant responses to Fe-stress. Table 2 also reveals that a considerable fraction of the carbon flowing through the glycolysis pathway is diverted to secondary metabolism, particularly flavonoid and isoprenoid biosynthesis, which increased at eCO_2_ under Fe-limited conditions in root tissues. Under Fe-stress, most proteins included in lignin biosynthesis, such as phenylalanine ammonia-lyase and peroxidase, were downregulated in the roots of soybean plants. Lignin provides mechanical strength to the plant’s secondary cell walls, which protect cells from abiotic stresses and serve as the structures that first perceive and respond to environmental stresses. Hence, the downregulation of phenylpropanoid biosynthesis (Appendix A) could save carbon and energy for other metabolic processes [32].

The capacity of cellular redox regulation is crucial to maintaining the activity of many physiological processes [33]. Oxidative stress is often neutralized by enzymatic and non-enzymatic antioxidative systems [34]. The expression of glutathione S-transferase proteins was higher in soybean plants exposed to Fe-limitation (Table 2 and Table 3). Glutathione S-transferases are involved in several plant functions as detoxification of xenobiotics, secondary metabolism, growth and development, tetrapyrrole metabolism, and against biotic and abiotic stresses [35,36]. Höhner et al. [37] showed that Fe deficiency increased the expression levels of glutathione S-transferases in *Chlamydomonas reinhardtii*. Similarly, Fe deficiency-induced changes in the protein profile of Arabidopsis thaliana roots and glutathione S-transferases were considered highly expressed proteins [30]. Moreover, the interaction of eCO_2_ and Fe-limitation induced the ascorbate-glutathione pathway in roots. Among the antioxidant defense mechanisms, the ascorbate-glutathione pathway is crucial in mitigating further damage to soybean plants caused by reactive oxygen species and derivatives produced during metabolic activity.

Plant growth and development occur due to the global balance between protein synthesis and degradation [33]. Our proteomic analysis showed that many enzymes involved in protein synthesis were downregulated by eCO_2_, under Fe-sufficient and Fe-limited conditions, in root and leaf tissues. These results could be associated with the fact that increased levels of carbohydrates can affect gene expression through their role as signaling molecules. Sugars could be involved in photosynthetic acclimation, whereby the additional carbohydrates formed under eCO_2_ conditions might cause the downregulation of photosynthetic gene transcripts and suppress protein synthesis [17]. However, photosynthetic acclimation does not always completely negate the positive results eCO_2_ has on plant growth. The positive effects of eCO_2_ on plant growth are well studied, but the role of hormone pathways in regulating the growth responses under eCO_2_ is slightly understood. Gibberellins are a class of diterpenoid hormones involved in several growths and developmental processes, including stem elongation, leaf expansion, flower development, and germination [38]. Biosynthesis of gibberellins includes 2-oxoglutarate/Fe(II)-dependent dioxygenases that are upregulated at eCO_2_, under Fe-limited and Fe-sufficient conditions, in this study. An increase in gibberellins expression under eCO_2_ has also been reported in species such as *Ginkgo biloba* [39], *Arabidopsis thaliana* [40], and *Populus tomentosa* [41]. Our finding suggests that eCO_2_ might play a role in signaling, allowing higher plant growth rates.

Most enzymes involved in the stress responses, including heat-shock proteins, had low expression levels under Fe-limitation in soybean plants. The role of heat-shock proteins is to manage protein folding and promote cellular protection, protein homeostasis, and cell survival against several environmental and metabolic stresses [14,42]. We infer that soybean plants had lower levels of protein structure protection under Fe-limited conditions. In addition, proteins involved in cell-wall organization, including UDP-glucose 6-dehydrogenase, cellulose synthase, and xyloglucan endotransglucosylase, were downregulated by Fe-stress and by the interaction of eCO_2_ with Fe-stress in root tissues (Appendix A). The reduced level of cell wall-modifying genes could limit cell expansion as plant growth decreased under Fe-limitation. The extent and severity of abiotic stresses or the crops selected are crucial in determining the effects of eCO_2_ and Fe-stress. Therefore, eCO_2_ and Fe-stress influence the growth and yield of plants and their subsequent adaptation to future climate changes and should require more attention from the scientific community.

## 4. Materials and Methods

### 4.1. Plant Material and Growth Conditions

A previously identified highly-CO_2_ responsive soybean variety Wisconsin Black [43], was used as plant material. This study was performed at the Biotechnology School of Catholica University (Portugal). Plants were grown under hydroponic conditions in black plastic pots (5 L) filled with an aerated, full-strength nutrient solution with the following composition: 1.2 mM KNO_3_, 0.8 mM Ca(NO_3_)_2_, 0.3 mM MgSO_4_.7H_2_O, 0.2 mM NH_4_H_2_PO_4_, 25 μM CaCl_2_, 25 μM H_3_BO_3_, 0.5 μM MnSO_4_, 2 μM ZnSO_4_.H_2_O, 0.5 μM CuSO_4_.H_2_O, 0.5 μM MoO_3_, 0.1 μM NiSO_4_, and 20 μM Fe(III)-EDDHA. The solution was buffered by MES (1mM, pH 5.5) addition and changed every 3 d. The growth chamber was controlled to maintain the temperature at 25 °C (day period) and 20 °C (dark period), relative humidity at 75 %, and photosynthetic photon flux density at 325 μmol s^−1^ m^−2^ (daytime light). After 7 days of pre-treatment in the complete nutrient solution under aCO_2_, plants were transferred to a nutrient solution with Fe(III)-EDDHA at 0.5 μM (Fe-limited) or 20 μM (Fe-sufficient). Then, half of the plants were grown at 400 ppm (aCO_2_), and the other half were grown at 800 ppm (eCO_2_) in independent growth chambers for 12 days. The CO_2_ concentration was continuously monitored and maintained by an automated CO_2_ control system, which measured and adjusted the CO_2_ concentration from soybean planting to the end of the experiment.

### 4.2. Evaluation of Plant Biomass

At the end of the experiment, plants were dried in an oven at 70 ºC until constant weight for total plant biomass determination.

### 4.3. Leaf Gas Exchange Parameters

Leaf gas exchange measurements were performed on day 17 after the beginning of the experiment. We randomly selected the first expanded trifoliate leaf from 3 plants (*n* = 9), and the photosynthetic rate was measured using a portable photosynthesis system (LI-6400XT; LI-COR, Lincoln, NE, USA). The CO_2_ in the leaf chamber was set to match the CO_2_ treatment, with a photosynthetic photon flux density of 500 μmol photon m^−2^ s^−1^ at 25 °C. Moreover, the transpiration rate and stomatal conductance were determined.

### 4.4. Root and Leaf Carbohydrates

Root and leaf samples (*n* = 5) were evaluated for carbohydrate analysis at the end of the experiment. The extraction protocol was described by López-Millán et al. [44]. About 100 mg of plant material was grounded using liquid nitrogen, suspended in 5 mM H2SO4, vortexed for 30 s, and then boiled for 30 min. Samples were centrifuged at 2320× *g* for 10 min, supernatant was filtered through a 0.45 mm PTFE filter, and the volume was adjusted to 2 mL and stored at −80 °C until further analysis. The HPLC system consisted of an ion exchange aminex HPX-87H Column (Bio-Rad, Hercules, CA, USA) maintained at 40 °C. The mobile phase was 5 mM H_2_SO_4_ at a flow rate of 0.6 mL min^−1^.

### 4.5. Protein Extraction and LC−MS/MS Analysis

The protein extraction was based on the protocol developed by Wu et al. [45]. We used three biological replicates to perform the proteome analysis. From each sample, 250 mg of frozen plant tissue was ground in liquid nitrogen. The powdered tissue was dissolved in cold acetone (−20 °C) with TCA (10% wt/vol), homogenized with acid-washed sand, and centrifuged at 15,000× *g* for 5 min at 4 °C to collect the precipitated proteins. The pellet was resuspended in cold TCA/acetone, vortexed, and centrifuged at 15,000× *g* for 5 min at 4 °C to collect proteins and repeated until the pellet turned white. Then, the pellet was resuspended twice in 1.5 mL of cold acetone, centrifuged at 15,000g for 5 min at 4 °C, and the supernatant was discarded. Next, the pellet was air-dried in a fume hood and resuspended in an SDS extraction buffer for 1.5 h. Then, centrifuged at 15,000× *g* for 10 min, the supernatant was collected into new Eppendorf tubes, mixed with an equal volume of Tris-buffered phenol, and vortexed for 3 min. The mixture was centrifuged at 15,000× *g* for 5 min, and the lower phenol phase was collected into a new Eppendorf tube. An equal volume of washing buffer I was added to the phenol phase, vortexed by 3 min, and centrifuged at 15,000× *g* for 5 min at room temperature. The upper phase (phenol phase) was collected into a new Eppendorf tube, and 0.1 M ammonium acetate in methanol was added to a final volume of 2.0 mL. The mixture was vortexed for 30 s and stored at −20 °C for 2 h to precipitate the phenol-extracted proteins. The solution was centrifuged at 15,000× *g* for 10 min at 4 °C, and the supernatant was discarded. The protein pellets were resuspended with 0.1 M ammonium acetate in methanol and centrifuged at 15,000× *g* for 5 min at 4 °C, and the supernatant was rejected. Therefore, the pellets were resuspended in 80% (*v*/*v*) acetone, centrifuged at 15,000× *g* for 5 min at 4 °C, and the supernatant was removed. The protein pellets were air-dried at room temperature and resuspended in 100 μL of rehydration buffer (50 mM ammonium carbonate, 8 M urea). The concentrations of the protein extracts were determined by the Bradford assay [46].

Each sample was processed for proteomic analysis following the solid-phase-enhanced sample-preparation (SP3) protocol and enzymatically digested with Trypsin/LysC as previously described [47]. Protein identification and quantitation were performed by nanoLC-MS/MS following an already published procedure [48] using a nanoLC flow rate of 300 nL/min. For protein identification and quantification, the UniProt database was considered for the Glycine max Proteome (2020_01). The proteomics data analysis was performed with the Proteome Discoverer 2.4.0.305 software (Thermo Scientific, San Jose, CA, USA.

### 4.6. Database Search and Protein Quantification

Only high-confidence peptides and proteins with at least two unique peptides detected in all three replicates were used in quantification. For all quantified proteins, only those showing a fold change of above 1.2 or below 1/1.2 (for the mean of the three replicates) in the quantitative ratios and *p* < 0.05 were considered as differentially expressed proteins (DEPs). Protein sequences were submitted to the online Mercator 4 annotation tool [49] for proteome annotation. Protein functions were categorized using Mapman bin codes, and the protein abundance ratio was visualized through MapMan software [50]. Pathway enrichment analysis of DEPs was performed using Fisher’s exact test. Information about the subcellular location was accomplished from the location available from SUBA4 [51].

### 4.7. Statistical Analysis

Data were analyzed using SPSS software (SPSS version 26.0). Analysis of variance (2way ANOVA) was used to determine differences among different treatments after data normality and equal variance analysis. The means ± SE were calculated for each parameter.

## 5. Conclusions

Elevated CO_2_ and Fe-stress had profound effects on plant biomass, sugar content, and Pn, with interactive effects of eCO_2_ and Fe-stress on plant biomass. Overall, the CO_2_-induced increase in biomass was not significantly different between Fe+ELE and Fe-ELE treatments. We performed proteomic analysis to analyze DEPs affected by the interaction of Fe-limitation and eCO_2_ in soybean plants. Overall, root and leaf tissues contained 705 and 589 DEPs, respectively. Based on pathway enrichment analysis, cell wall organization, glutathione metabolism, photosynthesis, stress-related proteins, and biosynthesis of secondary compounds changed in roots to cope with Fe-stress. Moreover, the enhanced plant growth by eCO_2_ supplied with sufficient or insufficient Fe was associated with the increased abundance of proteins involved in glycolysis, starch and sucrose metabolism, biosynthesis of gibberellins, and decreased levels of protein biosynthesis. The understanding of plant productivity and adaptation to future climate changes will also improve through future studies created to compare responses across various cultivars and during different periods of stress exposure.

## Figures and Tables

**Figure 1 ijms-23-13632-f001:**
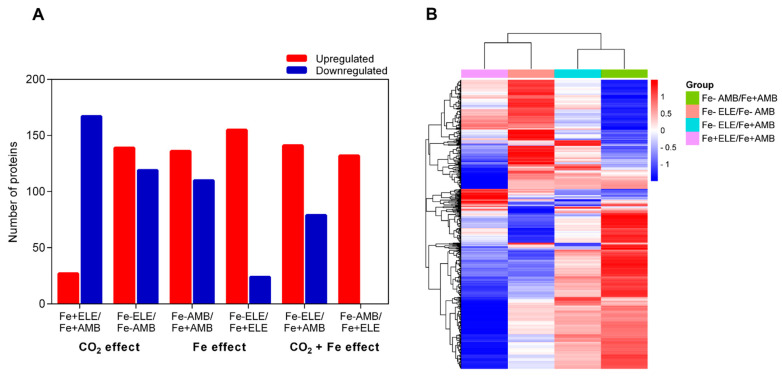
Soybean root proteome profiles under eCO_2_ and Fe-limitation. (**A**) The number of upregulated and downregulated proteins in soybean root in response to eCO_2_ and Fe-limitation; (**B**) Cluster analysis of all differently regulated proteins among different treatments. Fe+AMB, Fe-sufficient + aCO_2_; Fe+ELE, Fe-sufficient + eCO_2_; Fe-AMB, Fe-limitation + aCO_2_; Fe-ELE, Fe-limitation + eCO_2_.

**Figure 2 ijms-23-13632-f002:**
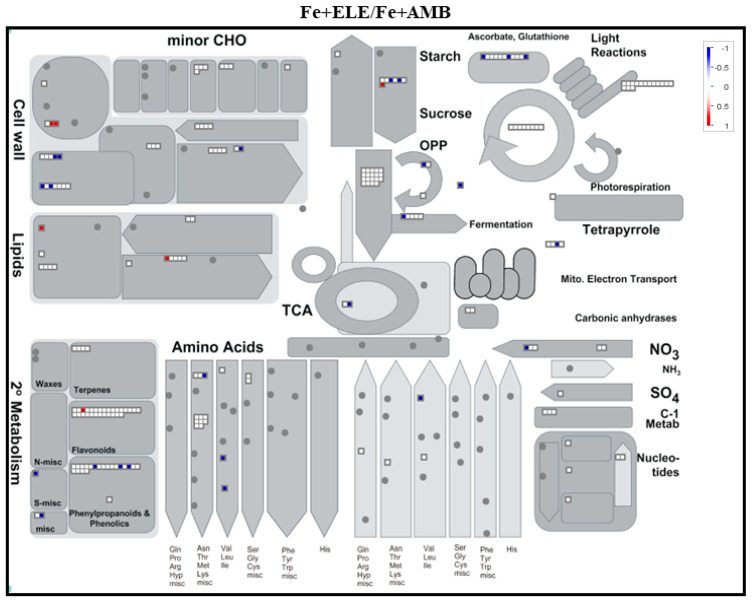
MapMan metabolism overview maps showing changes in DEPs in root tissues under eCO_2_ and Fe-limitation. Fe+AMB, Fe-sufficient + aCO_2_; Fe+ELE, Fe-sufficient + eCO_2_; Fe-AMB, Fe-limitation + aCO_2_; Fe-ELE, Fe-limitation + eCO_2_. Boxes represent log2 expression values, genes in red are upregulated, and those in blue are repressed.

**Figure 3 ijms-23-13632-f003:**
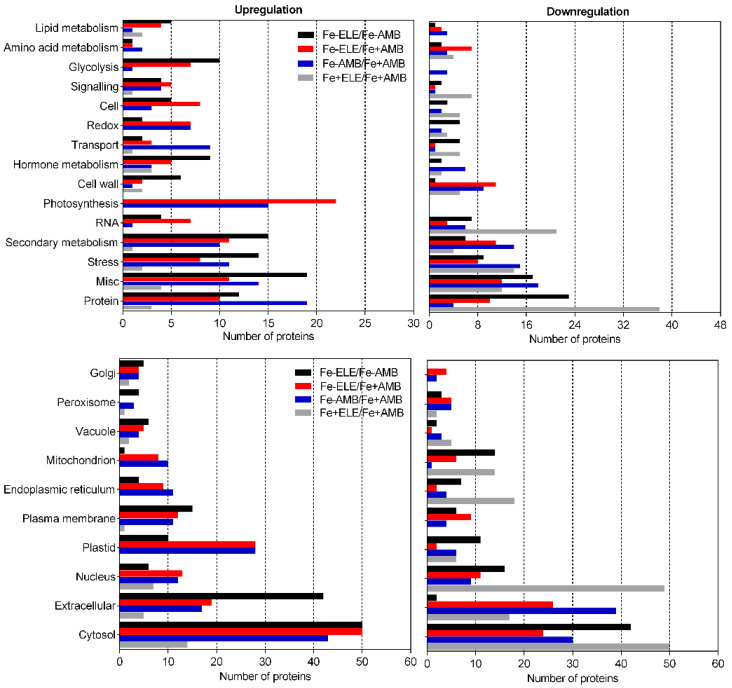
Numbers of DEPs identified from soybean roots at different CO_2_ levels under sufficient and limited Fe-supply according to functional categories and subcellular compartments by MapMan. Fe+AMB, Fe-sufficient + aCO_2_; Fe+ELE, Fe-sufficient + eCO_2_; Fe-AMB, Fe-limitation + aCO_2_; Fe-ELE, Fe-limitation + eCO_2_.

**Figure 4 ijms-23-13632-f004:**
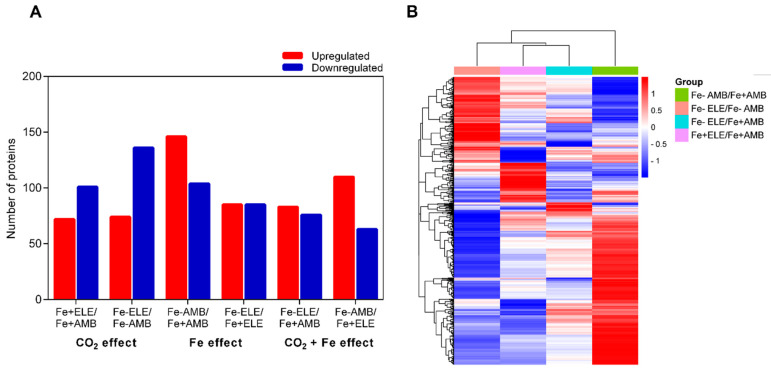
Soybean leaf proteome profiles under eCO_2_ and Fe-limitation. (**A**) The number of upregulated and downregulated proteins in soybean leaf in response to eCO_2_ and Fe-limitation; (**B**) Cluster analysis of all differently regulated proteins among different treatments. Fe+AMB, Fe-sufficient + aCO_2_; Fe+ELE, Fe-sufficient + eCO_2_; Fe-AMB, Fe-limitation + aCO_2_; Fe-ELE, Fe-limitation + eCO_2_.

**Figure 5 ijms-23-13632-f005:**
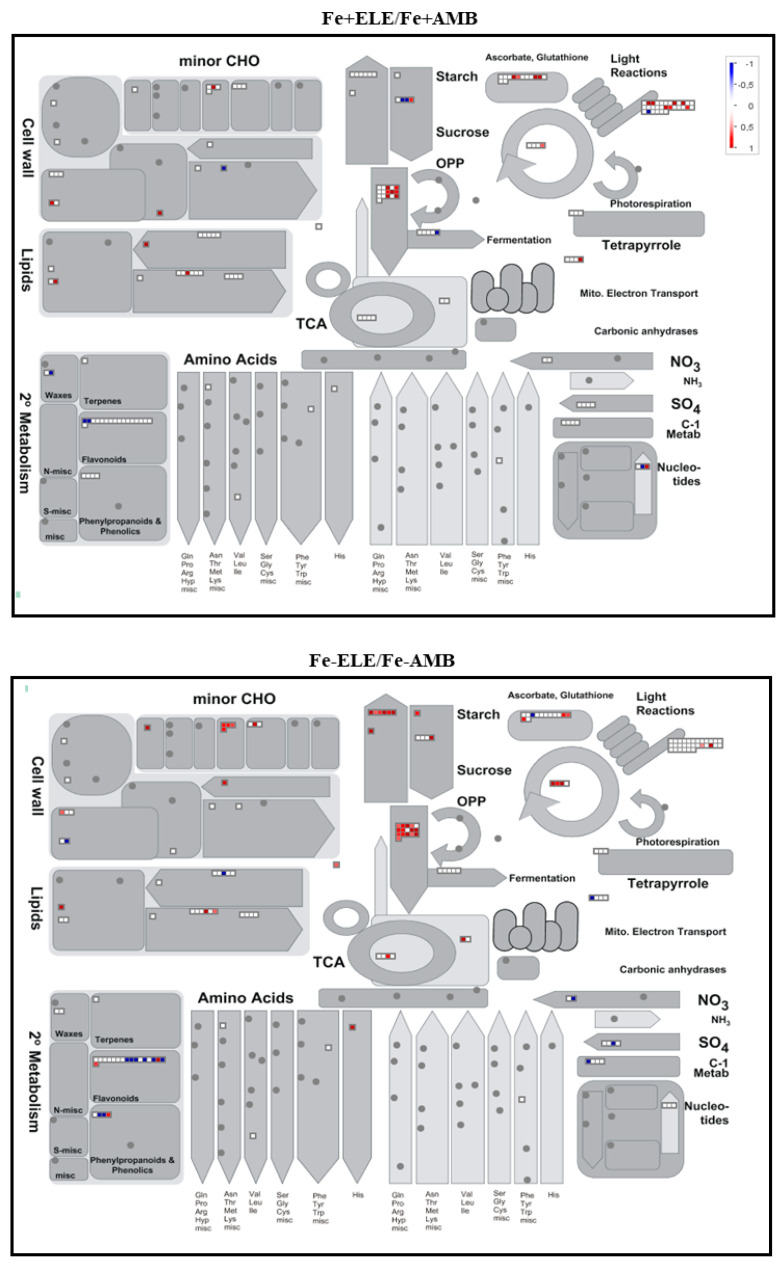
MapMan metabolism overview maps showing changes in DEPs in leaf tissues under eCO_2_ and Fe-limitation. Fe+AMB, Fe-sufficient + aCO_2_; Fe+ELE, Fe-sufficient + eCO_2_; Fe-AMB, Fe-limitation + aCO_2_; Fe-ELE, Fe-limitation + eCO_2_. Squares represent log2 expression values, and genes in red are upregulated, and those in blue are repressed.

**Figure 6 ijms-23-13632-f006:**
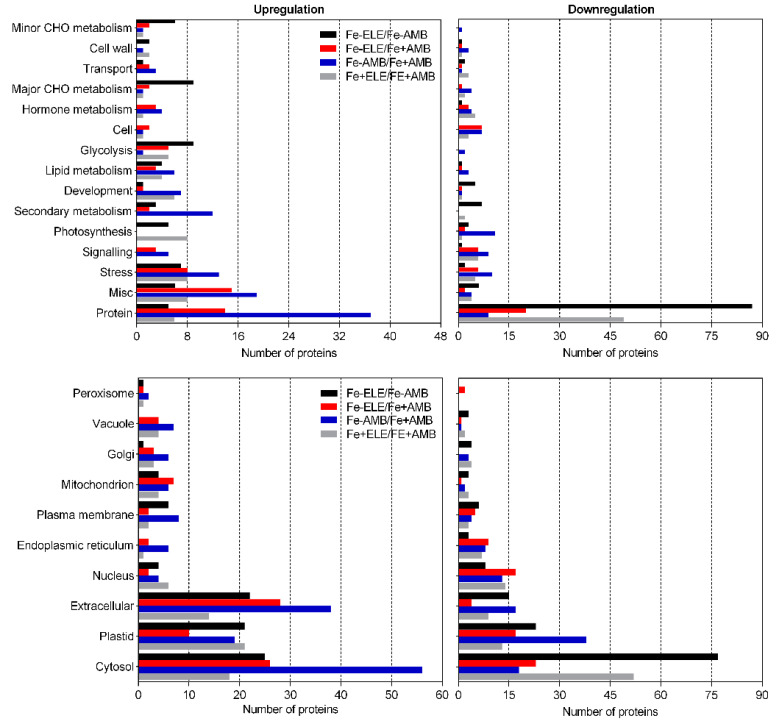
Numbers of DEPs identified from soybean leaves at different CO_2_ levels under sufficient and limited Fe-supply according to functional categories and subcellular compartments by MapMan. Fe+AMB, Fe-sufficient + aCO_2_; Fe+ELE, Fe-sufficient + eCO_2_; Fe-AMB, Fe-limitation + aCO_2_; Fe-ELE, Fe-limitation + eCO_2_.

**Table 1 ijms-23-13632-t001:** Effects of eCO_2_ and Fe-limitation on biomass, sugar content, and gas exchange parameters in soybean plants.

	Treatments				Two-Way ANOVA			
Measurements	Fe+ELE	Fe+AMB	Fe-ELE	Fe-AMB	d.f.	CO_2_	Fe	CO_2_ × Fe
Total biomass (g plant^−1^)	3.14 ± 0.19 ^a^	2.14 ± 0.15 ^b^	2.91 ± 0.12 ^a^	1.18 ± 0.08 ^c^	1	**94.29; <** **0.001**	**18.04; <** **0.001**	**6.59; 0.021**
Pn (µmol m^−2^ s^−1^)	11.21 ± 0.43 ^a^	11.43 ± 0.55 ^a^	9.21 ± 0.40 ^b^	8.90 ± 0.80 ^b^	1	0.006; 0.939	**15.17; <** **0.001**	0.21; 0.654
gs (mol m^−2^ s^−1^)	0.13 ± 0.01 ^a^	0.37 ± 0.03 ^b^	0.12 ± 0.01 ^a^	0.30 ± 0.02 ^c^	1	**85.74; <** **0.001**	**6.568; 0.021**	**5.42; 0.033**
Tr (mol m^−2^ s^−1^)	2.46 ± 0.19 ^a^	5.22 ± 0.36 ^b^	2.28 ± 0.16 ^a^	4.13 ± 0.29 ^d^	1	**54.19; <** **0.001**	3.84; 0.068	4.44; 0.051
Sugar (roots, μmol g FW^−1^)	26.17 ± 3.72 ^a^	10.92 ± 1.12 ^b^	19.89 ± 2.36 ^c^	8.75 ± 0.92 ^b^	1	**9.90; 0.009**	4.82; 0.051	1.85; 0.201
Sugar (leaves, μmol g FW^−1^)	52.22 ± 4.03 ^a^	33.62 ± 1.21 ^b^	46.70 ± 2.54 ^a^	34.80 ± 2.44 ^b^	1	**37.71; <** **0.001**	0.761; 0.406	1.82; 0.210

Note: Fe+AMB, Fe-sufficient + aCO_2_; Fe+ELE, Fe-sufficient + eCO_2_; Fe-AMB, Fe-limitation + aCO_2_; Fe-ELE, Fe-limitation + eCO_2_; Data are mean ± SE (n = 5). Different lowercase superscript letters in the same row indicate the significant difference at p < 0.05 level. CO_2_, Fe, and CO_2_ × Fe indicate CO_2_ treatment, Fe treatment, and the interaction of CO_2_ by Fe treatment, respectively. Results from the analysis of variance with degrees of freedom (d.f.), F ratios, and probabilities (p) for some parameters. Significant effects are shown in boldface.

**Table 2 ijms-23-13632-t002:** Pathway enrichment analysis of DEPS in roots of soybean plants under eCO_2_ and Fe-limitation.

Bin Code	Bin Name	Fe+ELE vs. Fe+AMB	*p*-Value	Fe-AMB vs. Fe+AMB	*p*-Value	Fe-ELE vs. Fe+AMB	*p*-Value	Fe-ELE vs. Fe-AMB	*p*-Value
1.1.1	PS. Light reaction. Photosystem II	-	-	UP	2.9 × 10^−5^	UP	1.7 × 10^−5^	-	-
1.1.2	PS. Light reaction. Photosystem I	-	-	UP	1.5 × 10^−2^	UP	3.8 × 10^−5^	-	-
1.3	PS. Calvin cycle	-	-	-	-	UP	4.4 × 10^−4^	-	-
4.1	Glycolysis. Cytosolic branch	-	-	-	-	UP	1.5 × 10^−3^	UP	1.7 × 10^−2^
10	Cell wall	-	-	DOWN	5.6 × 10^−7^	DOWN	1.6 × 10^−3^	-	-
13.1.3.4	Amino acid metabolism. Synthesis. Aspartate family. Methionine	-	-	-	-	DOWN	6.9 × 10^−3^	-	-
15.2	Metal handling. Binding, chelation, and storage	-	-	DOWN	3.3 × 10^−3^	DOWN	4.3 × 10^−7^	-	-
16.1	Secondary metabolism. Isoprenoids	-	-	DOWN	8.3 × 10^−3^			UP	1.6 × 10^−4^
16.2.1	Secondary metabolism. Phenylpropanoids. Lignin biosynthesis	-	-	DOWN	1.8 × 10^−3^	DOWN	2.6 × 10^−2^	-	-
16.8	Secondary metabolism. Flavonoids	-	-	DOWN	1.4 × 10^−2^	-	-	UP	3.1 × 10^−4^
17.6	Hormone metabolism. Gibberellin	UP	1.4 × 10^−4^	DOWN	8.3 × 10^−3^	-	-	UP	3.6 × 10^−4^
20.1	Stress. Biotic	-	-	DOWN	5.6 × 10^−3^	-	-	UP	2.1 × 10^−3^
20.2.1	Stress. Abiotic. Heat	-	-	DOWN	4.6 × 10^−2^	-	-	-	-
21.2.1.3	Redox. Ascorbate and glutathione. Ascorbate. l-galactose-1-phosphate phosphatase	-	-	-	-	UP	2.1 × 10^−2^	-	-
26.9	Misc. Glutathione S-transferases	-	-	UP	1.5 × 10^−2^	-	-	DOWN	1.3 × 10^−3^
26.12	Misc. Peroxidases	-	-	DOWN	2.7 × 10^−3^	-	-	UP	1.3 × 10^−3^
26.13	Misc. Acid and other phosphatases	-	-	DOWN	6.1 × 10^−3^	-	-	UP	3.6 × 10^−4^
26.14	Misc. Oxygenase	-	-	DOWN	4.2 × 10^−2^	-	-	-	-
27.3	RNA. Regulation of transcription	DOWN	9.4 × 10^−3^	-	-	-	-	-	-
27.4	RNA.RNA binding	DOWN	4.4 × 10^−2^	-	-	-	-	-	-
29.2.1	Protein. Synthesis. Ribosomal protein	DOWN	4.2 × 10^−2^	UP	2.9 × 10^−5^	-	-	DOWN	1.0 × 10^−14^
34	Transport	-	-	UP	3.1 × 10^−2^	-	-	DOWN	3.7 × 10^−2^

**Table 3 ijms-23-13632-t003:** Pathway enrichment analysis in leaves of soybean plants under eCO_2_ and Fe-limitation.

Bin Code	Bin Name	Fe+ELE vs. Fe+AMB	*p*-Value	Fe-AMB vs. Fe+AMB	*p*-Value	Fe-ELE vs. Fe+AMB	*p*-Value	Fe-ELE vs. Fe-AMB	*p*-Value
1.1.1	PS. Light reaction. Photosystem II	-	-	DOWN	1.9 × 10^−4^	-	-	-	-
1.1.4	PS. Light reaction. Photosystem I	-	-	DOWN	3.0 × 10^−2^	-	-	-	-
2.1	Major CHO metabolism. Synthesis	-	-	DOWN	1.0 × 10^−2^	-	-	UP	1.2 × 10^−4^
3.4	Minor CHO metabolism. Myo-inositol	-	-	-	-	-	-	UP	2.0 × 10^−2^
4.1	Glycolysis. Cytosolic branch	UP	2.0 × 10^−3^	-	-	-	-	UP	2.0 × 10^−2^
11.9.3.3	Lipid metabolism. Lipid degradation. Lysophospholipases. Glycerophosphodiester phosphodiesterase	-	-	DOWN	1.7 × 10^−3^	-	-	UP	3.5 × 10^−3^
13.2.4	Amino acid metabolism. Degradation. Branched chain	-	-	UP	3.3 × 10^−2^	-	-	-	-
16.8	Secondary metabolism. Flavonoids	-	-	UP	6.1 × 10^−4^	-	-	-	-
20.1	Stress. Biotic	UP	6.1 × 10^−3^	UP	3.7 × 10^−4^	UP	1.6 × 10^−7^	-	-
20.2.1	Stress. Abiotic. Heat	-	-	DOWN	2.7 × 10^−3^	-	-	-	-
26.9	Misc. Glutathione S-transferases	-	-	UP	3.8 × 10^−2^	-	-	-	-
26.12	Misc. Peroxidases	-	-	UP	1.3 × 10^−4^	UP	1.8 × 10^−5^	-	-
28.1.3	DNA. Synthesis/chromatin structure. Histone	DOWN	1.5 × 10^−2^	UP	4.1 × 10^−2^	-	-	DOWN	2.4 × 10^−2^
29.2.1	Protein. Synthesis. Ribosomal protein	DOWN	2.8 × 10^−14^	UP	1.9 × 10^−13^	-	-	DOWN	9.5 × 10^−14^
29.2.2	Protein. Synthesis. Ribosome biogenesis	DOWN	9.8 × 10^−3^	-	-	-	-	DOWN	3.8 × 10^−3^
29.3.4	Protein. Targeting. Secretory pathway	-	-	-	-	-	-	DOWN	2.8 × 10^−3^
29.5.1	Protein. Degradation. Subtilases	-	-	-	-	-	-	UP	2.3 × 10^−3^
29.5.3	Protein. Degradation. Cysteine protease	-	-	UP	2.5 × 10^−3^	UP	4.1 × 10^−3^	-	-
30.5	Signaling. G-proteins	-	-	-	-	DOWN	4.2 × 10^−2^	-	-
33.1	Development. Storage proteins	UP	6.5 × 10^−6^	UP	8.3 × 10^−3^	-	-	DOWN	1.4 × 10^−2^
34.9	Transport. Metabolite transporters at the mitochondrial membrane	DOWN	7.8 × 10^−3^	-	-	-	-	-	-

## Data Availability

The data presented in this study are available in the Appendix A.

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
