# Peer review of "Effect of the Interaction between Elevated Carbon Dioxide and Iron Limitation on Proteomic Profiling of Soybean"

_ijms, 2022, doi:10.3390/ijms232113632_

Round 1

Reviewer 1 Report

It is an important piece of research work and authors justified with properplanning and execution. 

Author Response

nothing to answer to reviewer 1 

Reviewer 2 Report

The manuscript is well organized and clearly descripted. Nevertheless, there are some problems should be revised. See below.

L94. “Figure 0”?

L98. In table 1, some words are bond, whereas some are not. Blank is missing after “;”.

L114. In figure 1A, the x-axis is number, not differentially expressed proteins. In figure 1B, different meanings are indicated using same word “Group”.

L134. In figure 2, some words are too small and hardly to be read.

L143. In figure 3, take lipid metabolism as an example, word should be uppercase the first letter.

L160. In figure 4, same problem as in figure 2.

L179. In figure 5, same problem as in figure 3.

L196. In figure 6, same problem as in figure 1.

L204. In table 2, as “E” is used to indicate the number, using superscript is wrong.

L404. No blank between number and °C, which is the only for number and unit.

Author Response

Response to Reviewer 2 Comments

Point 1: L94. “Figure 0”?

Response 1: The problem with “figure 0” was solved and the sentence was changed accordingly (L95)

Point 2: L98. In table 1, some words are bond, whereas some are not. Blank is missing after “;”.

Response 2: The issues that occurred in Table 1 were resolved (L101).

Point 3: L114. In figure 1A, the x-axis is number, not differentially expressed proteins. In figure 1B, different meanings are indicated using same word “Group”.

Response 3: The figure was changed, and the y-axis title was changed to “number of proteins”, and in the figure 1B one of the words “group” was deleted (L118).

Point 4: L134. In figure 2, some words are too small and hardly to be read.

Response 4: The Figure 2 was changed for a better interpretation and readability (L144).

Point 5: L143. In figure 3, take lipid metabolism as an example, word should be uppercase the first letter.

Response 5: The figure 3 was adjusted as suggested by the reviewer (L154).

Point 6: L160. In figure 4, same problem as in figure 2.

Response 6: The figure 4 was changed as suggested by the reviewer (L172).

Point 7: L179. In figure 5, same problem as in figure 3.

Response 7: The figure 5 was changed as suggested by the reviewer (L193).

Point 8: L196. In figure 6, same problem as in figure 1.

Response 8: The figure 6 was changed as suggested by the reviewer (L210).

Point 9: L204. In table 2, as “E” is used to indicate the number, using superscript is wrong.

Response 9: In tables 2 and 3 the superscript was removed (L232).

Point 10: L404. No blank between number and °C, which is the only for number and unit.

Response 10: The blank between the number and ºC was deleted as suggested by the reviewer throughout the manuscript (L388, 398 and 413).

Reviewer 3 Report

Line 63-68 Please check the accuracy of the results from reference [13]. “The increase in biomass was higher in Fe- limited plants compared to Fe-sufficient plants” This statement seems doubtful.

Line 68-71 Please provide the premises of this piece of result from reference [14], under what conditions did these proteins differentially accumulated?

Line 75-79 Do you mean these proteins are differentially accumulated among species under different Fe or CO2 conditions? Please check the accuracy of the statement.

In your Figure 1, why not show the results of Fe-ELE/Fe+ELE? To my understanding, in Fe-ELE/Fe+AMB comparison, two factors are considered, while other comparison groups, only one factor is considered.

Line 120, “705 proteins were differentially expressed in root tissues”? What factor did you use to obtain the DEP number in root tissues, same with the DEP number in leaf tissues, please clarify.

Line 379-380, how did you implement the CO2 treatment, were there four groups of materials with Fe and CO2 treatment combinations being used for proteomic profiling? Please describe this part clearly.

In your research, only young plants were used, it would be better if you can include more measurement data from mature plants and compare results from more verities and more developmental stages.

Author Response

Response to Reviewer 3 Comments

Point 1: Line 63-68 Please check the accuracy of the results from reference [13]. “The increase in biomass was higher in Fe- limited plants compared to Fe-sufficient plants” This statement seems doubtful.

Response 1: The statement “The increase in biomass was higher in Fe-limited plants compared to Fe-sufficient plants” was related to the reference number 5 and not to reference 13 as incorrectly suggested in our manuscript. Our intention was to highlight a higher percentage increase in biomass in Fe-limited plants compared to Fe-sufficient plants. Therefore, the statement was rephrased in L63-66.

Point 2: Line 68-71 Please provide the premises of this piece of result from reference [14], under what conditions did these proteins differentially accumulated?

Response 2: The proteins differentially accumulated under Fe deficiency as described in L69-70.

Point 3: Line 75-79 Do you mean these proteins are differentially accumulated among species under different Fe or CO2 conditions? Please check the accuracy of the statement.

Response 3: We wanted to describe only proteins accumulated under Fe-limited conditions from reference number 16. Thus, we made a new paragraph with the statement “Therefore, we consider that it is relevant to discover the complex response mechanisms of soybean plants to future climatic conditions, such as eCO2 and Fe-limitation” to highlight the importance of eCO2 and Fe-limitation (topics of this research) in our manuscript (L80-84).

Point 4: In your Figure 1, why not show the results of Fe-ELE/Fe+ELE? To my understanding, in Fe-ELE/Fe+AMB comparison, two factors are considered, while other comparison groups, only one factor is considered.

Response 4: We have analyzed 6 combination ratios (Fe+ELE/Fe+AMB, Fe-AMB/Fe+AMB, Fe-ELE/Fe+AMB, Fe-ELE/Fe-AMB, Fe-AMB/Fe+ELE, Fe-ELE/Fe+ELE) as demonstrated in Table S3 and S4. The main objective is to study the CO2 effect (Fe+ELE/Fe+AMB and Fe-ELE/Fe-AMB), the Fe effect (Fe-AMB/Fe+AMB), and the CO2 and Fe interaction (Fe-ELE/Fe+AMB) relative to control plants (Fe+AMB). However, we changed Figures 1 and 4 to include the other combinations we didn’t have in the previous version of the manuscript.

Point 5: Line 120, “705 proteins were differentially expressed in root tissues”? What factor did you use to obtain the DEP number in root tissues, same with the DEP number in leaf tissues, please clarify.

Response 5: The factor used to obtain DEPs number was a fold change ratio of above 1.2 or below 1/1.2 and p < 0.05, as described in L124-125 and L167-168.

Point 6: Line 379-380, how did you implement the CO2 treatment, were there four groups of materials with Fe and CO2 treatment combinations being used for proteomic profiling? Please describe this part clearly.

Response 6: The section 4.1. related to growth conditions was totally rephrased to better explain the variables (CO2 and Fe) studied in this study (L380-396)

Point 7: In your research, only young plants were used, it would be better if you can include more measurement data from mature plants and compare results from more verities and more developmental stages

Response 7: We no longer have measurement data from mature plants and varieties or even further stages of development. As suggested by the reviewer, this could be useful data, but at the same time, it can be difficult to compare all the suggested variables. The plants were chlorotic at the end of the experiment (3 weeks) due to Fe limitation and would possibly not reach maturity. Furthermore, this manuscript was the last experiment of my Ph.D. thesis that is about to end.